# Anticorruption, Cultural Norms, and Implications for the APUNCAC

**Danny Singh** 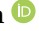

Department of Humanities and Social Sciences, Teesside University, Middlesbrough TS1 3BX, UK; D.Singh@tees.ac.uk

**Abstract:** Corruption is a phenomenon that has received global attention from academics, policy makers and international donors. Corruption may be defined as the abuse of power for private gain. Activities include bribery, extortion, rent-seeking behaviour, cronyism, patronage, nepotism, embezzlement, graft and engagement with criminal enterprises. However, patronage, nepotism and gift giving are frequently viewed in many Asian and African cultures as acceptable practices that promote efficiency and smooth relationships. This article examines these practices in contexts including Afghanistan, Papua New Guinea, Russia, China and South Asia, discusses various rationales for these practices, and seeks to understand how these practices can be reconciled with international efforts to combat corruption. This article focuses on the implications with regard to the Anticorruption Protocol to the United Nations Convention against Corruption (APUNCAC) and the proposal to establish a body of United Nations (UN) inspectors to investigate charges of corruption and refer cases to dedicated domestic anticorruption courts. This article suggests that UN inspectors and international norms against corruption are not incompatible with traditional cultural practices. This article draws upon the experiences of Hong Kong and Singapore, where corruption was endemic, to demonstrate that local cultural norms can be rapidly changed when independent inspectors are established and receive support from institutions that are free from manipulation by domestic authorities.

**Keywords:** corruption; international public law; governance; rule of law; culture

## 1. Introduction

Corruption hinders the rule of law and democracy, results in human rights abuses and economic stagnation and permits organised crime and support for terrorism and other major threats to human security (Annan 2020). The preamble of the United Nations Convention against Corruption (UNCAC) recognises that corruption undermines the security and stability of a country and hinders institutions, democracy, sustainable development, justice and the rule of law (United Nations Office on Drugs and Crime (UNODC) 2004, p. 5). Local and cultural traditions may, however, undermine Western definitions of corruption. If gift giving, nepotism, patronage and the blurring of public and private spheres are part of, for example, the Asian social and cultural fabric, how should the international community respond?

Despite traditional practices that involve gift giving and the exchange of favours, the establishment of institutions to fight corruption is, arguably, not irreconcilable with the existence of local norms that seemingly validate corrupt practices. This may be illustrated through the experiences of Hong Kong and Singapore. Both experienced endemic corruption but achieved rapid success in establishing clean governments after the introduction of independent anticorruption inspectors. The key to this success was the insulation of inspectors from manipulation by political leaders.

The Anticorruption Protocol to the United Nations Convention against Corruption (APUNCAC) is a 200 page draft treaty that intends to establish an independent body of

United Nations (UN) inspectors and courts dedicated to anticorruption. This proposal aims to fill legislative gaps to: fight money laundering and reveal the identity of beneficial owners; investigate and prosecute the crime of obstruction of justice; permit independent investigations into accusations of corruption and abuse of power; and transfer cases to national anticorruption courts (Yeh 2021, p. 2). However, it is possible that the APUN-CAC might be viewed as inconsistent with traditional Asian cultural practices involving patronage, gift giving, *guanxi*, *blat* and *bakhshish*. This article focuses on a key question, *how can* the *APUNCAC be reconciled with local cultural norms that reinforce practices defined by Westerners as corruption?*

Section 2 of this article defines corruption and explores the concept of corruption. Section 3 introduces concepts including traditional patronage, gift giving, *guanxi*, *blat* and *bakhshish* from Asian contexts that, from a Western perspective, appear to involve corruption. Section 4 describes the experiences of Singapore and Hong Kong, which experienced endemic corruption that was effectively addressed through the introduction of independent anticorruption inspectors. Section 5 addresses the primary question—is the APUNCAC, a proposed treaty that would introduce UN anticorruption inspectors, a strategy that can be reconciled with traditional patronage, gift giving, *guanxi*, *blat* and *bakhshish* practices? Section 6 concludes.

## 2. What Is Corruption?

The term "corruption" derives from a Latin term, "corruptus," meaning to break, ruin, spoil or contaminate (Nicholls et al. 2006, p. 1). This notion has implications with respect to morality and ethics. In Western democracies, corruption is generally viewed as immoral, evil, theft, or abuse of power. These terms may be used interchangeably. Corruption may be defined narrowly as the abuse of authority for private advantage (Kurer 2015, p. 31). According to this definition, there are a variety of activities that qualify as corruption. For instance, von Alemann (1989, p. 858) states that corruption is frequently related to bribery, which covers the abuse of authority and power for individual gain but does not have to be financial. This definition, involving the abuse of authority for personal gain, has been adopted by a variety of international anticorruption agencies (World Bank 1997; United States Agency for International Development (USAID); Boucher et al. 2007; Transparency International 2021). Clark (1993, p. 8) expanded the definition of corruption to include the intent of public officials to abuse official positions for private gain. The intent of a public official to attain a personal advantage through abuse of authority would be defined as corruption even if the gain is not achieved.

Political and social scientists have sought to expand the definition of corruption. In politics, corruption concerns a public official, the public, and a recipient of the favourable treatment by the public official, such as a private group or third party who is affected by the activity of that office (Philp 2008, p. 315). The anticipated and actual beneficiary may be the same. Berta van Schoor (2017, p. 15) defines corruption as an illegal deal involving collective goods transferred between actors within public or private domains that are then transformed into private payoffs. Nye (1967, p. 417) and Khan (1996, p. 12) define corruption as the deviation of behaviour from formal responsibilities of an elective or appointed public position, due to private (personal, familial or clique) wealth, power or status advantage. The potential exchange of material privileges and/or benefits, involving private payoffs, is a form of clientelism that favours close family (i.e., nepotism) or a preferred ethnic or tribal group (Heidenheimer and Johnston 2002, p. 6). A system of clientelism results in elites and higher classes leveraging their control to reallocate state resources to loyal supporters (King 2009, p. 29). This clientelist exchange may conform to legal fictions but is an abuse of elite power. Hence, corruption involves some form of private advantage that includes a public official abusing his (or her) post or duties for monetary gain or political support (Mény and Rhodes 1997, p. 98). Discovering the varied forms of corruption and revealing this type of illicit activity is a difficult task (Rayner 2012, p. 111). This is due, in part, to customary social, cultural and business practices that,

in certain parts of the world, facilitate and promote what may otherwise be defined as corruption.

While popular perceptions of what qualifies as corruption usually focus on bribery, corruption may encompass a variety of activities linked with criminal behaviour and/or other forms of malfeasance. There is a distinction between low-level and high-level corruption—what may be delineated as grand corruption.

Low-level corruption usually includes bribery. Bribery comprises interactions between the public and civil servants. High-level corruption involves state leaders, parliamentarians and other senior officials (Morris 2011, pp. 10–11). Bribery may be defined as an illicit payment made to a public official linked to an obligatory action. Without the payment, the act, service or favour would not be possible (Davis 2019, p. 88). The bribe is offered in exchange for an action by the public official (Pieth et al. 2007, p. 111). For example, a service can be delayed or denied by a civil servant, until a bribe is paid (Chan 2005, p. 109). Administrative corruption also includes extortion (inducement of a bribe), gift giving (a gift provided by a beneficiary or employee in expectation of a service or favour), kickbacks (illegal payment projected as a reimbursement for special treatment or other categories of improper services obtained), graft (diversion of public funds for private purposes), embezzlement (theft of state resources and funds) and fraud (forming a false company, claiming salaries from ghost employees, and overcharging expenses). Favouritism is a type of administrative corruption that includes nepotism, cronyism and patronage.

According to Rose-Ackerman (2008, pp. 330–31), there are four components of low-level corruption that are facilitated by discretionary powers exercised by government officials. First, the practices of *purchasing posts* concerns candidates who are willing to pay for a position for the benefit. Second, *facilitation payments* to a few senior officials can include the allocation of a limited supply of construction permits to the highest bidders. Third, *paying to speed up a public service or transaction* involves the payment of a bribe from the public, as beneficiaries or applicants. Fourth, *turning a blind eye* includes public officials and a police force that chooses to overlook illicit goods and driving offences.

Grand corruption is when a bureaucracy and state resources are tapped by ruling political elites that fosters corruption from the top to bottom levels (Maged 2012, p. 357). Rose-Ackerman (2008, p. 331) identifies three facets of grand corruption. First, *the entirety or part of the public sector generates conditions for bribery* that is practised in the higher and lower administrative levels. Second, there is a *lack of transparency or control of governmental spending* that violates spending limits and subverts funding regulations. Third, a *government undertakes contracts with private firms* for profitable mining, oil, and construction deals to create a system to collect kickbacks and lower taxes on the private agency.

Therefore, typologies of corruption are helpful to identify several practices of administrative and grand corruption that includes political elites, public officials and law enforcement actors that are engaged in the pursuit of their private and/or political advantage. The consequences of corruption include citizen alienation and erosion of state and political legitimacy that exacerbates volatility and social fragmentation (Nye 2009, p. 288).

However, functionalists believe that all features of a society serve a purpose and are essential for a society's survival. Based on this sociological theory, functionalists would argue that the causes of corruption relate to a political apparatus which permits bribery to grease the wheels of a bureaucracy to reduce red tape (Merton 1968). An overly rigid bureaucracy and rules create a system where the public suffers delayed services. This creates opportunities where bribery is expected or demanded as the price of expeditious service delivery. A regulated bureaucracy with extensive rules and service delays facilitates and promotes opportunities for bribery in exchange for political favours from public officials.

### 3. The Social Construction of Corruption: Non-Western Contestation

Favouritism is part of everyday practices in many parts of the world. Favouritism occurs when public officials use their position of power to benefit close family, friends

and preferred people linked to the distribution of jobs and state resources, hindering meritocracy (Nongogo 2006, p. 7). There are several types of favouritism, including nepotism, cronyism and patronage.

Nepotism is the practice of appointing relatives into a position, or the provision of a promotion or other benefit for a close relative. This can include an increase in pay, promotions and privileges to retain loyalty and respect to family and may be related to business and politics (Dilts et al. 1985, p. 66). Cronyism is the activity of an office holder to prefer, reward and/or award trusted colleagues, which may include low-skilled friends, or other ties with preferential treatment to a position or to benefit from a contract without bidding (Khatri 2011, p. 62). Patronage occurs when a politician unequally distributes resources, jobs, or contracts in anticipation of political loyalty (Stokes 2009, p. 650). The politician's attempt to expand his or her votes can include passing legislation from clients to create a network for the attainment of resources and other benefits for their receivers (Kenny 2017, p. 52). Nepotism, cronyism and patronage involve the misuse of state resources and funds, and function as a "corruption complex" (Olivier de Sardan 1999, p. 27). Olivier de Sardan (1999) employs the term 'corruption complex' to refer to acts involving nepotism, abuse of power, embezzlement and various forms of misappropriation, traffic of influence, prevarication, and abuse of the public purse.

Therefore, corruption is not simply a moral wrong or rule breaking behaviour. Instead, corruption may be perceived as part of the cultural fabric in many societies. Olivier de Sardan (1999) recognises that corruption is cultural because kinship, tribal and village ties are preferred circles for well-paid jobs, privileges and opportunities to retain morally acceptable relations in many African societies. This is referred to as the "moral economy" because a value system can be culturally engrained, morally accepted and entrenched from the top to bottom levels (Olivier de Sardan 1999, pp. 25–26). The expression 'moral economy' serves to pinpoint certain social norms common in today's Africa, which communicate and influence the practices of corruption. Within a moral economy, clientelism is unavoidable due to political elites distributing state resources to family, friends, local tribes and political factions so that patrons can retain their clients.

Gift giving is another practice that is part of the moral economy. In Asia, gift giving is one of the customary practices and cultural norms that are not perceived as corruption (Gezgin 2009, pp. 111–12). Expensive handouts or tips can be offered by a tipper in exchange for a debt or anticipated favour to cement long-term relationships (Singh 2020, p. 22). Small gifts can also be provided to form short-term relationships in exchange for an immediate favour, goods or services from the receiver (Singh 2020, p. 22). For example, a gift can be a payment, alcoholic beverage or a free service pass (Asher 2007, p. 168). Hence, the gift establishes a reciprocal relationship because the return is compulsory, and this gives the gift-giver power. The gift-giver creates the obligatory gesture to induce a public official to attend to their duties more rapidly (Olivier de Sardan 2015, p. 40). The corruption complex comprises moral obligations of favouritism. The reciprocal obligation of gift giving functions as part of the social and cultural fabric in many non-Western societies. Gift giving involves a variety of practices.

In China, *guanxi* is the provision of gifts to establish an obligatory gesture or familiarity between two parties because refusing a gift results in a "loss of face" (Yang 1989, p. 42). Although the social practices of gift giving assist private material advantage, the dissemination of small gifts between neighbours and families preserves positive relations (Yang 1994, pp. 80–81, 140). For this reason, Ast (2019, p. 107) asserts that *guanxi* does not involve a definite payment for a service that does not comply with the traditional definition of bribery. In addition, *guanxi* can be utilised in everyday working relations. Yang (1994) describes a scenario when a worker presented a gift to their manager for a few days of authorised absence. Although the manager initially declined the gift, the employee gave the gift in the presence of all colleagues. In this situation, the manager honoured the employee's leave demand due to other colleagues witnessing the acceptance of the gift (Yang 1994). Hence, gift giving forms a social obligation and contract between two

parties and is engrained in Chinese cultural values. Once a gift is accepted, a favour must be fulfilled to avoid loss of face and to maintain positive relations. Perito and Kristoff (2010, p. 2) suggest that gift giving is deemed corruption or an unethical practice in Western democratic states, but this activity is perceived as good manners in many Asian and African societies. Corruption remains a main challenge throughout Asia. According to the Global Corruption Barometer—Asia, 74 per cent of its 20,000 survey contributors in 17 Asian countries perceive that government corruption is a huge problem in their state, and almost one-fifth paid a bribe for a public service within the last 12 months (Transparency International 2020, pp. 10, 20–21). In Asian contexts, the desire to preserve social harmony and promote business loyalty may increase the likelihood of corruption, but it would be incorrect to infer that all Asian culture is susceptible to corruption. Informal channels to obtain favours, goods and/or services in a legitimate fashion can be common in many societies to preserve strong relations among parties involved.

Informal contacts help to form networks of solidarity and provide benefits for those who are connected. In the context of Russia, informal networks that deviate from formal processes are needed to obtain limited goods and services. These informal procedures are referred to as *blat*. *Blat* helps to maintain circles and support mutual responsibilities, personal and friendship ties. It is a social support network that establishes and maintains moral duty to help others (Ledeneva 1998, pp. 40–41). In this system, bosses allocate favours to preserve their altruistic and positive self-image. Generally, no money is exchanged through these informal channels, but in some cases people are paid-off (Ledeneva 2017, p. 38). Similar to *guanxi*, *blat* is used to form and maintain long-lasting personal relationships so that authority figures can retain their reputable self-image and disseminate favours. Both practices in Russia and China focus on creating, and maintaining, personal and amicable relationships between connections and parties.

In a variety of South Asian contexts, informal practices, which may be deemed as corrupt practices, frequently occur in rural villages. In India, Gupta (1995) describes a scenario in which an official provides a public service from his private house rather than a public office. This situation blurs lines between public and private functions. Money is exchanged informally for the service. This means that a house can function for public services. Gupta (1995, p. 379) gives the specific example of an Indian land official, named Sharmaji, who held approximately 5000 plans of land registry within his home and demanded money from villagers to register or alter land titles. The villagers did not contest paying bribes or that the house was used to perform official duties, but complained about their difficulty in contacting state officials. In the absence of contact with them, Sharmaji's services were necessary and accessible. Hence, the incompetence of Sharmaji's service was the problem cited by local villagers rather than a private space being utilised for public duties by means of paying bribes. If the latter is not disputed, then the blurring of public and private spheres and bribery remain as acceptable behaviour in some non-Western societies. Corruption is thus socially constructed to given cultures.

In another South Asian context, Ruud (1998) recounts the social construction of bribery and corruption in the case of Bengali villagers. In one scenario, Kalo held adequate qualifications to attain a local hospital post but was competing against several hundred candidates to fill 30 posts. Kalo understood that he required informal channels to influence his application with contacts and bribery at the top level (Ruud 1998, p. 13). Kalo bribed a senior hospital clerk, who was also his neighbour, to pay well-placed recruiters and provide a cut for the clerk. Although Kalo's cousin's in-laws were friends with the Town Commissioner, they did not have significant influence to issue an endorsement letter (Ruud 1998, p. 14). Consequently, Kalo's relatives did not proceed with payment because he was afraid of adversely affecting the friendship. In this context, Kalo did not contest the payment of bribes to his personal connections, who were friends, to also payoff distant appointers to attain a recommendation letter (Ruud 1998, p. 17). This case demonstrates that the use of bribery and paying off personal contacts to gain favour, such as attaining a public post, are culturally understood as informal procedures rather than corruption.

Favouritism is frequent in South Asia and can be correlated to relations with political figures. During the 1970s in Kathmandu, Nepal, rickshaw wallahs were encouraged by a Cabinet Minister to purchase tempos (auto rickshaws) with a long-term interest-free bank loan that were only granted if the loanee was related to the Minister or affiliates of the Minister's loyal political backers (Kondos 1987, p. 17). Bank loans were provided on conditions of favouritism. Favouritism is engrained as part of everyday Nepalese practice. Based on this premise, the unsuccessful loan applicants did not regard this as corruption, nor complained about the methods used by others to obtain the loans.

When considering these South Asian contexts, Sharmaji used his home as a private space to conduct public responsibilities with regard to land titles and amendments, which blurred the public and private spheres. Kalo relied on nepotism and a bribe to try to obtain a hospital job. Moreover, rickshaw wallahs formed favourable relations with the Cabinet Minister and close political supporters. Blurring public-private domains, nepotism, bribery and favouritism remain part of commonplace practices and the cultural fabric of obtaining services, goods and appointments in South Asia. Favouritism can be acceptable, and the cultural norm of gift giving remains embedded as a common feature of everyday relations rather than being deemed as corruption.

The perceptions of the socially constructed practices of what does and does not constitute corruption, even if culturally engrained as everyday relations, can undermine trust in public services and the state. For instance, a study based on theology and ethics by Pavarala and Malik (2010, p. 20) contends that corruption can expand past gift giving and in some cultures, womanising and lying are deemed as corrupt practices.

Studies regarding public perceptions of corruption in Papua New Guinea and Afghanistan focus on commonplace interactions with public officials and illicit activities that are deemed immoral. A study conducted by Walton (2012) in nine Papua New Guinea provinces with a total of 1825 survey participants showed that corruption is associated with the abuse of public authority for private gain (28 per cent), evil activities (26 per cent) and all immoral practices (17 per cent). When questions were asked about damage to society, the most frequently reported was minor immoral acts such as young women soliciting sexual services for money (68 per cent). Other activities such as bribery (60 per cent) and embezzlement (39 per cent) were perceived as fully corrupt but were deemed as practices that were less morally harmful (Walton 2012, pp. 3–7). Therefore, most of the recipients believed that moral and legal coercions caused harmful and corrupt practice.

Similarly, corrupt activities damage moral obligations in Afghanistan. Despite this premise, corrupt acts are deemed as unavoidable when engaging with Afghan public officials. The term *fasad* is roughly defined as a dirty act, which includes prostitution, and is culturally detested (De Lauri 2013, p. 534). Bribery is termed *rishwat* or *bakhshish* in courts. While corruption is viewed as immoral, Afghan civilians, according to a United States Institute of Peace (USIP) publication, believe that bribery is inescapable when interacting with public officials, such as civil servants and judges, who receive payoffs to avert delays in service (Barfield 2012, p. x). This relationship is embedded in everyday Afghan interactions with public servants, but such activities impede morality. A nationwide survey conducted by Integrity Watch Afghanistan (IWA) (2018) with 8130 civilians across 34 provinces asked numerous questions regarding civilian attitudes and experiences of corruption, how corruption impacts on public services, opinions of the government, and efforts to fight corruption. Most respondents (39 per cent) deemed corrupt people as sinful; this was followed by the "venal" motivation of bribery (23 per cent); and that individuals disbursing bribes are sinful (43 per cent) which made the respondents feel sad (21 per cent) (Integrity Watch Afghanistan (IWA) 2018, p. 45). Even if corruption, namely bribery or gift giving, is deemed inevitable, it is still perceived as harmful by local Afghanis, promoting social inequality, impeding the delivery of public services, and intensifying negative perceptions of the state.

Although gift giving, bribery and favouritism, in the form of patronage, *guanxi*, *blat*, or *bakhshish*, may be present in many cultures to preserve working and social patronage

relations in exchange for benefits or posts, these activities damage the social contract between the state and its citizens. Even when corruption is practised, people may not complain due to the rules of the game. However, these practices are still deemed as a social harm or an evil in society (as evident in Papua New Guinea and Afghanistan). Within a political economy analysis, informal channels and formalised bribery alienate disfavoured groups and impoverished people in society. Corruption and patronage may be part of cultural fabric and function as a means of retaining small long-lasting circles and business transactions, but these practices promote the public's perception that the state is illegitimate, distrust of the public sector, and the view that the social contract has been breached. The existence of patronage, gift giving, *guanxi*, *blat*, and *bakhshish* are not inconsistent with international anticorruption norms. The experiences of Singapore and Hong Kong offer positive case studies suggesting how even engrained corruption can be overcome.

## 4. Success Stories in Singapore and Hong Kong

Singapore and Hong Kong were formerly British Crown colonies. During the colonial period, these city-states were highly corrupt, and corruption was part of everyday life. Britain recognised the need to fight corruption in Singapore and Hong Kong. After initial failures, strong British governors imposed draconian solutions to address, and ultimately solve, the problem of corruption.

Singapore was a British Crown colony from 1946 until 1963. Despite high levels of police corruption, in December 1937, the British Crown colony established an Anti-Corruption Branch (ACB) of the Criminal Investigation Department within the Singaporean Police Force, charged with the responsibility for fighting corruption (Quah 2017, p. 9). Unsurprisingly, the ACB failed to combat entrenched police corruption. On 27 October 1951, an investigation team found that three police detectives and a gang of robbers stole a cargo of opium worth United States (US) $130,330 (Quah 2017, p. 9). Due to this shortcoming, the ACB was succeeded by an independent anticorruption agency. In October 1952, the British assisted with the creation of the Corrupt Practices Investigation Bureau (CPIB) as an autonomous body. The Bureau was detached from the police to combat rampant corruption in the national police force and investigate cases of corruption (Root 1997, p. 46). The CPIB holds extensive legal powers with the authority to investigate and prosecute cases of corruption within public and private departments (Organization for Economic Co-Operation and Development (OECD) 2013, pp. 59–61).

As part of the commitment to anticorruption, the June 1959 People's Action Party (PAP), a pro-independence party, tried to curtail corruption. Corruption was particularly rampant in the civil service, resulting in PAP's general election success in May 1959. Subsequently, on 17 June 1960, the Prevention of Corruption Act (PCA) was passed which expanded the powers of the CPIB. The PCA specifically contained 32 sections and defined corruption as varieties of gratification. Moreover, the PCA introduced a custodial sentence of five years and a hefty fine in cases of corruption (Quah 1995, p. 395). If an investigated person was found guilty of illegal gratification, then they had to pay back the amount obtained as a bribe (Quah 1995, p. 395). The PCA provided the CPIB with arrest powers, the right to search individuals who are arrested, and authorisation to investigate bank accounts involved in crime. In 1963, the PCA was revised to provide CPIB officers the authority to summon and examine witnesses. Significantly, the PCA ensured that the mere intention of receiving a bribe, even if not physically obtained, could constitute grounds for conviction. Moreover, Singapore citizens could be prosecuted for corruption if activities were committed outside of the country (Quah 1995, p. 396).[1] In addition to the PCA and amendments that provided the CPIB with more authoritative investigation and arrest

---

[1] Singapore eventually attained sovereign independence on 9 August 1965.

powers, the 1989 Corruption (Confiscation of Benefits) Act was established to confiscate private advantage derived from corrupt activity, including from a deceased offender.[2]

Singapore reformed the public administration to encourage meritocracy and restructure pay levels to ensure that wages were competitive with the private sector (Quah 1994, pp. 210–11). This was to avoid brain drain from the public sector to the private sector. In March 1972, the salaries of civil servants were increased to compete with the private sector and increments ensued throughout the 1970s and 1980s that made senior civil servants earn higher wages than the United States Federal Service (Quah 1995, pp. 398–99). Furthermore, the police could not interfere with the CPIB because, unlike the British models of anticorruption agencies, its status as an anticorruption agency remained separate and independent from the police (Quah 1995, pp. 401, 406). To reduce corruption in vice areas and field investigation, Singaporean police officers were switched to other sectors every three years (Quah 2006, p. 63). Police officers had their salaries raised and training was centred on educating them in police integrity and ethics (Pyman et al. 2012, p. 66). The political will of consecutive Singaporean governments helped to raise the integrity and ethics of police officers and challenge corruption (Jennet 2007, p. 2). These changes, and incessant vows by the Singaporean government, have reinforced the capabilities of the CPIB. Since 1995, the CPIB has adopted the use of polygraph testing (Corrupt Practices Investigation Bureau 2020). Hence, the CPIB established its independence. These changes, reinforced by vigorous rotation of police, made Singapore one of the least corrupt countries in the world (Saxena and Bagai 2010, pp. 16–17).

Hong Kong was resistant to many of the reforms that were successful in Singapore. The reasons for this were that it lacked a sense of national pride, lacked the political will of a prime minister and was inhibited by its history as a colony of China (Kuan 1981, p. 40). From 1842 to 1941, corruption in Hong Kong was a regular customary practice of the middle and lower governmental tiers (Manion 2004, p. 28). In the 1950s, public services expanded. During this time, officials demanded payments from those wishing to be favoured for housing distribution or construction projects. In addition, popular corrupt practices of smuggling and bribery were prevalent when dealing with mainland Chinese officials (Grantham 1965, p. 121).

Throughout the 1960s and 1970s, the British Crown colony was plagued by an embedded commission economy, where public officials expect a payment to fulfill a service or to provide protection. A shocking example involved a poor hawker who failed to pay five dollars each day as "tea money" to corrupt officials engaged with the Wanchai corruption syndicate, and had to surrender his youngest daughter (Independent Commission against Corruption 2005, p. 8). Hong Kong Chinese[3] in non-official posts offered gifts to get things done via a connection or *guanxi* as a "cultural ploy" (Chan 2005, p. 96). Those in power distributed favours, that would later be called upon, which in turn became accepted daily rituals to pay "tea money" to government officials (Chan 2005, p. 97). These social interactions were inherited from the Chinese custom of paying "convenience money" to reward hard work. Bribery became an additional tax to enable working-class people, such as cab drivers and hawkers, to survive (Chan 2005, p. 97).

During this period, the police were also engaged in syndicated corruption to protect illicit businesses by extorting money to ignore their compulsory responsibilities (Manion 2004, p. 30). Syndicated corruption comprised organised criminal syndicates that permeated the public sector, including law enforcement agencies and the criminal justice system (Man-wai 2013, p. 99). A taxi driver could purchase a monthly sticker-label from a syndicate to guarantee immunity from traffic prosecution (Man-wai 2013, p. 99). Ordinary Chinese found it fruitless, foolish and hazardous to question or confront government officials (Manion 2004, pp. 30–31). In other words, corruption was customary for the majority of middle-aged and older Hong Kong Chinese whose experience of the mainland

---

2   Corruption (Confiscation of Benefits) Act, Republic of Singapore. No. 16, Cap 65A (Acts Supplement, 31 March 1989), 2000 Rev. Ed. (1 July 2000).

3   Ethnic Chinese residents now constitute approximately 92 per cent of Hong Kong's ethnic composition (Central Intelligence Agency World Factbook 2020).

was a regime notorious for corruption. This experience shaped "folklore" regarding the bureaucracy of Hong Kong, where corruption was accepted as an ordinary way of life and citizens lost faith in public administration (Lethbridge 1985, p. 14). Hong Kong Chinese officials believed that Hong Kong governmental workers would also be corrupt. This belief bred corruption and namely bribery, due to the perception that corruption was an ordinary practice when dealing with a government department (Manion 2004, p. 31). Therefore, the culture of corruption was customarily embedded and prevalent within Hong Kong. Corruption operated as a form of everyday interactions and relations and thus shared similarities with tipping, the protection of small circles and regular bribery in China, Russia and Afghanistan (*guanxi*, *blat* and *rishwat*). Even if corruption was deemed as a social problem, it was engrained as part of a Hong Kong culture that was inherited from China. Corruption was culturally embedded and rife. Combating it became a seemingly impossible task in Hong Kong.

In Hong Kong, the government made a strong commitment to combat systemic corruption and reinstate public confidence in the government (Quah 1995, p. 403). The Anti-Corruption Branch (ACB) was created in 1952 as an attached special unit of the Royal Hong Kong Police Force (RHKPF) criminal investigation department to investigate bribery and other cases related to corruption (Lo 2009, p. 124).

Due to prevalent organised police corruption, referred to as syndicated corruption, in 1971, the ACB separated and was upgraded from the criminal investigation department to the Anti-Corruption Office (ACO) (Quah 2013, p. 233). The ACO reviewed the previous Prevention of Corruption Ordinance to create the Prevention of Bribery Ordinance (PBO) that serves as the main source of domestic anticorruption law. The PBO provided the government with the power to prosecute civil servants for corruption if they failed to provide an adequate clarification for maintaining a high living standard (Quah 1995, p. 401). Moreover, the Attorney-General could inspect bank accounts and search for other information. The PBO was terminated when a British Chief Superintendent, Peter Godber, was caught in a sensational scandal.

ACO officers found evidence of financial resources surpassing US $780,000 and placed immigration on high alert to detain Godber if he attempted to flee Hong Kong (Manion 2004, p. 33). On 8 June 1973, Godber evaded immigration and fled to Britain. This enraged the Hong Kong public, triggered university student protests and undermined the credibility of the ACO (Independent Commission against Corruption 2005, p. 8; Quah 2013, p. 233). Student demonstrations specifically chanted the slogan "Fight Corruption, Catch Godber" (Chan 2005, p. 97). This scandal resulted in the Blair-Kerr Commission of Inquiry to investigate Godber's escape and the inefficiency of the PBO (Scott 2011, p. 404). The Commission wanted the complete separation of the ACO from the RHKPF and a newly established anticorruption agency that would retain independence from the police force to circumvent further sheltering of senior police officers like Godber (Lethbridge 1985, p. 98). The Governor, Sir Murray MacLehose, could not ignore that corruption was a prevalent social problem in Hong Kong. Corruption existed in the police force and several other departments, which is why people lacked faith in the ACO and decided that it needed to be replaced (Manion 2004, p. 34). In 1974, Godber was extradited from Britain to face trial resulting in a conviction for accepting bribes and conspiracy (Carroll 2007, p. 174; Scott 2011, p. 404). Subsequently, Godber served a four-year jail sentence. Corruption was recognised as a serious social problem in Hong Kong that British colonial governors could not ignore. Further measures were taken with the development of an independent anticorruption agency that recognised that the police department could be a contributor to corruption.

On 15 February 1974, the Independent Commission against Corruption (ICAC) was established to stamp out corruption and restore confidence in the government (Quah 1995, p. 403). In its early administration, in December 1974, the ICAC was criticised for relying on police manpower, namely 58 officers. A year later, the ICAC continued to be deployed in its Operations Department, but the Commission knew little about police culture and

the nature of policing duties (Lethbridge 1985, pp. 108–9). The disgrace of Godber still lingered, and many doubted the rigour of renewed anticorruption efforts.

However, the ICAC remained committed and worked with the Police Corruption Prevention Group to fight police corruption in the 1970s (Bayley and Perito 2011, p. 13). Subsequently, the ICAC called for British support. Britain initially supplied nine officers, followed by 34 in 1975. These officers had no nepotistic, kinship or social networks entangled in Hong Kong Chinese culture (Lethbridge 1985, p. 110). In 1976, the ICAC waged a campaign against corruption syndicates and the Triad underworld. The operation reduced corruption within the police department and syndicates were unveiled. This resulted in the arrest of corrupt police officers, whilst music and massage parlours, brothels and dancehalls were closed due to the recession (Lear 1985, pp. 126, 128–29, 132). In October 1977, an estimated 140 police officers, ranging from constables to superintendents, from three Kowloon departments were arrested by the ICAC for engagement in syndicated corruption (Lear 1985, pp. 134–35, 142). British superintendents later pled for partial amnesty. Subsequently, it was recommended that the ICAC would not act upon complaints received about corruption prior to 1 January 1977, unless the accused was interviewed before that date (Lear 1985, p. 142). Public protests and riots were in the ascendancy which influenced Governor MacLehose to amend the Police Force Ordinance to include the summary dismissal of the Police Commissioner. This legislation functioned as a weapon to oust corruption at the highest level (Lear 1985, p. 144).

The ICAC retained independence because its staff, finance, structure and authority reported straight to a Governor that appointed the Commissioner (Quah 1995, p. 404). ICAC's personnel were appointed directly from the civil service. *The ICAC* intended to curtail widespread corruption within government departments through a threefold strategy related to "law enforcement, prevention and community education" (Independent Commission against Corruption 2007). This strategy focused on: disseminating information to communities about the legal definitions and ordinances of corruption; engaging officials in enforcement activity; encouraging citizens to report corruption; and intensifying societal disapproval of corruption (Manion 2004, p. 44). This approach enhanced the capacity of the ICAC to investigate and prosecute corrupt officials, educate citizens to encourage the reporting of corruption and increase the financial punishment for corrupt practices to decrease opportunities for engaging in corruption (Manion 2004, p. 27). ICAC officers had the authority to arrest when they suspected corruption and bribery offences, without a warrant, whilst obtaining the power to search sites to seize evidence for these offences (Quah 1995, p. 404). In short, the "three-pronged attack" provided the ICAC with a robust mandate to investigate, prevent and educate regarding corruption (Independent Commission against Corruption 2005, p. 12).

Moreover, in Hong Kong, many law enforcement officers owned properties or businesses across the border in China that made them susceptible to petty corruption from Chinese officials. As a response, the ICAC suggested that the Hong Kong government should frequently rotate police, immigration and custom officers stationed within border areas to reduce the misuse of their position for private benefit (Lo 1996, pp. 163–64). In June 2004, the ICAC investigated and arrested 20 people, which included five Organised Crime and Triad Bureau officers, for being paid "several thousand dollars" as bribes from the Wo Triad to protect their vice establishments (Lo 2009, p. 124).

High levels of public confidence were evident with the ICAC. An estimated 70 per cent of public opinion deemed ICAC's operations as effective or extremely effective and the level of corruption in the Hong Kong government was perceived to be low in the 1980s (Manion 2004, pp. 66, 83). Hong Kong inhabitants trusted the ICAC. The Commission's longer-term strategy was to intertwine traditional Hong Kong values with its own policies (Chan 2005, p. 104). The former British Crown colony established legal definitions, based on common-law, regarding advantage, bribery and corruption that were expressed through concepts and language that were culturally understandable, to change opinions on corruption from acceptable to intolerable behaviour (Chan 2005, p. 104). Therefore,

the ICAC concentrated on the broadcasting, naming and shaming of corruption. The Commission prosecuted corrupt police officers, supported by high-level political will, and a robust rotation strategy in a corrupt city-state that was changed into "a clean society" (Man-wai 2010, p. 138). Moreover, the culture of defined social harms and language of defined ordinances supported the beliefs, norms and values of ethnic Chinese residents. Hong Kong residents clearly knew that "advantage" covered the acceptance or offering of bribes with an intention to attain favours that was punishable under law due to decades of ICAC's effective "investigations and indictments" (Chan 2005, p. 105).

Although the ICAC was well-staffed, its anticorruption strategy was less focused on increasing the wages of civil servants to provide disincentives for corruption than the CPIB (Quah 1995, p. 406). The CPIB was understaffed, with 66 officers in comparison to ICAC's 1219 personnel, but relied on the support of a variety of governmental departments to assist with their duties (Quah 1995, p. 406). The CPIB created a model for independent anticorruption agencies that have since been established in other Asian countries to stamp out corruption (Quah 2013, p. 233). The CPIB operated secretly and did not communicate with the public, unlike the ICAC, which openly engaged with the public (Quah 1995, p. 407). The anticorruption strategy in Hong Kong focused on reducing corrupt opportunities, instead of combating incentives for engaging in corruption (Quah 1995, p. 409).

Despite this shortcoming, the ICAC was largely responsible for Hong Kong's rapid switch from widespread corruption to a clean government (Manion 2004, p. 27). Chinese Hong Kong culture shaped expectations of corruption. Regardless, corruption was recognised as a social problem. University students led protests to extradite Godber from England for prosecution in Hong Kong (Independent Commission against Corruption 2005, p. 8). The Godber case led to the establishment of the ICAC, which was supported by successive governments and promoted a strong commitment to curtail the culture of corruption. Both commissions in Singapore and Hong Kong retained independence from the police and the government to curb corruption, instead of relying on the police force (Quah 2013, p. 40). Both Prime Minister Lee Kuan Yew in Singapore and Governor MacLehose in Hong Kong supported the fight against corruption through the establishment of anticorruption laws and agencies (Quah 1995, p. 408). Preliminary funding from colonial Britain, strong political will from the consecutive governments, and ensuring that the agencies retained independence from the police helped to make both anticorruption agencies successful. Furthermore, both agencies supported legislation concerning corrupt activity or the intent of corruption and bribery, even if the suspect failed to attain private gain.

The preceding analysis suggests that the existence of local customs favouring corruption are not irreconcilable with international anticorruption norms. Corruption is widely understood to be a social problem, even when embedded as a seemingly permanent cultural practice. Strong efforts to curtail corruption in Hong Kong were successful in altering deeply embedded customary practices of corruption, where the payment of "tea money" bribes functioned to smooth interactions with Hong Kong officials and maintain illicit enterprises, syndicated corruption, and malfeasance.

The examples of Singapore and Hong Kong suggest why a global anticorruption strategy, involving international anticorruption inspectors, may be successful even when local culture appears to sanction embedded practices including patronage, gift giving, *guanxi*, *blat*, or *bakhshish*. International anticorruption norms embedded in the proposed Anticorruption Protocol to the UN Convention against Corruption (APUNCAC), involving the establishment of UN anticorruption inspectors and dedicated domestic anticorruption courts, are not irreconcilable with the existence of local customs and culture that promote corruption.

## 5. International Anticorruption Strategy

The United Nations Convention against Corruption (UNCAC), involving 187 States Parties, entered into force in December 2005. It functions as the only legally binding global anticorruption legislation. The Convention dedicates five sections to prevention, sanctions

and law enforcement, global cooperation, the recovery of assets to rightful owners and particular states, and the exchange of information (Hechler 2017, pp. 1–3). Moreover, the Convention includes definitions of the various practices of corruption, such as trading in influence, bribery, the abuse of authority and numerous acts of malfeasance (which specifically refers to bribery and the embezzlement of property) in the private sector (United Nations Office on Drugs and Crime (UNODC) 2004, pp. 17–19). The UNCAC is highly useful in framing policy dialogue at senior levels. It is regularly used to monitor channels of aid. The Implementation Review Group of the Conference of the States Parties to the UNCAC reviews anticorruption policies by inviting the participation of civil society and reviewers to conduct country visits and produce full country reports (Hechler 2017, pp. 3–4).

Despite the existence of the UNCAC and the creation of international law and norms to prohibit corruption, money laundering and the financing of terrorism remain. Weak investigation, weak prosecution, and the failure to seize the financial assets of organised crime syndicates and terrorists continue, due to weak provisions for tracing illicit proceeds of crime transmitted to offshore jurisdictions. While the provisions of the UNCAC frequently refer to "a court," "competent authority" or other appropriate means of arbitration, the UNCAC fails to create a body of UN inspectors to investigate charges of corruption and refer cases to dedicated domestic anticorruption courts. The relationship with domestic law appears overly broad and vague. There is a need to supplement the UNCAC with a clearer procedure for investigation, sanctions, the involvement of specific national courts, or integration with an ambitious global anticorruption court modelled on the International Criminal Court.

Yeh (2020) has, with determination, worked on the development of a 200 page draft document, creating the APUNCAC, to fill gaps that remain in international anticorruption law. The APUNCAC requires the beneficial owners of assets transmitted via the international financial system to disclose their ownership, with the objective of eradicating the anonymity of protecting the financial assets of criminals and terrorists (Yeh 2021, p. 2). The APUNCAC includes provisions that would enable UN anticorruption inspectors to investigate abuses of authority and efforts to impede justice. UN inspectors established by the APUNCAC would have the authority to transfer cases to national anticorruption courts. The APUNCAC would be established to reinforce international public law. All State Parties would be mandated to cooperate in the investigation and prosecution of corruption, including the extradition of suspects. The APUNCAC authorises UN inspectors to file reports of noncooperation. A failure of cooperation would be sanctioned through penalties levied by the World Bank and the International Monetary Fund, and exposed through reports published online by Transparency International (Yeh 2021, p. 2). The APUNCAC establishes procedures to vet the selection of judges and prosecutors employed by dedicated domestic anticorruption courts and monitor their performance. Vetting and monitoring would be performed by the UN Commission on Crime Prevention and Criminal Justice. This procedure would seek to ensure the meritocracy and accountability of the judiciary that serves domestic courts devoted solely to anticorruption (Yeh 2021, p. 15). Within this protected judicial system, independent UN inspectors would submit their investigative reports to dedicated national anticorruption courts.

An independent proposal by Mark Wolf, Senior Judge of the United States District Court for Massachusetts, would establish an international anticorruption court (IACC) modelled on the International Criminal Court (ICC) (Wolf 2018, pp. 145–46). Wolf proposes that the IACC would be led by impartial, competent international judges and staffed by an international corps of elite prosecutors. The IACC would apply the ICC principle of complementarity, i.e., domestic courts would have primary jurisdiction concerning crimes committed by corrupt political leaders in their states. This means that the IACC would only have the authority to investigate, prosecute, and sentence leaders and their collaborators for corruption as a last resort, if a nation is demonstrably unable or unwilling to prosecute (Wolf 2018, pp. 149–50).

Although Wolf's proposal has received global support from Colombia, Peru, the US and the UN, Stephenson (2016) has argued that an international court dedicated to punishing corruption is impossible because states that are ruled by corrupt elites, or sensitive to concerns regarding sovereignty, will not sign onto this proposal. The proposed international anticorruption court is ambitious because it would impose more controls, transparency and accountability on political leaders. This would pose a direct threat to the neopatrimonial political systems that many political elites utilise to survive. Yeh's proposed dedicated national anticorruption courts would also threaten corrupt systems of patronage. However, Yeh's proposal would presumably avoid the same level of concern that would be raised by the IACC, involving an international court that would function like the International Criminal Court, imposed by the international community as a condition of the UNCAC or World Trade Organisation (WTO) membership. Yeh's national anticorruption courts would avoid thorny issues of jurisdiction, operational definitions of behaviour that demonstrates unwillingness or inability to investigate and prosecute crimes of corruption, and the controversial proposal by Judge Wolf to impose the IACC as a condition of the UNCAC or as a condition of membership in the WTO.

## 6. Conclusions: Hope for the APUNCAC and Dedicated Domestic Courts

This article has focused its attention on Asian social, cultural, and business practices that appear to condone, facilitate and promote what Westerners would define as corruption. The experiences of Hong Kong and Singapore suggest that even entrenched corruption can be reversed in a relatively short period of time when independent investigators are installed.

The analysis in Section 3 refutes the notion that Western definitions of corruption are somehow inimical or irreconcilable with regard to local cultures involving patronage, gift giving, *guanxi*, *blat*, or *bakhshish*. While these practices may be embedded in the local social and cultural fabric that emerges through everyday business and social interactions, local actors understand that bribery is costly. It is tolerated not because Asians are somehow inherently more accepting of corruption, but because it seems impossible to fight the system.

Local citizens embrace clean government when it is made available to them. This is demonstrated by the experiences of Singapore and Hong Kong. Deeply entrenched corruption, involving the police, the judiciary, and all elements of government, was rapidly replaced by a culture of clean government when citizens were able to file complaints of corruption with truly independent anticorruption investigators who would take the complaints seriously and pursue them with vigour.

It would be counterproductive to dismiss the experiences of Singapore and Hong Kong as isolated instances involving special circumstances. Singapore and Hong Kong offer clear historical precedents regarding the path forward. What worked in Singapore and Hong Kong is what also works in modern Western democracies: independent investigators in combination with independent prosecutors and courts. The starting point is independent investigators because they serve to root out and expose corruption in the police and the judiciary. Their efforts can and should be reinforced by installing procedures to vet and monitor the performance of a dedicated corps of prosecutors and judges serving dedicated domestic anticorruption courts.

Yeh (2021) has described Ukraine's dedicated domestic anticorruption court and Ukraine's embrace of an international panel of anticorruption experts to vet the selection of judges to serve the dedicated domestic anticorruption court. This precedent suggests that dedicated domestic anticorruption courts, and international vetting of judges to serve those courts, may be embraced even in countries where corruption has reached the highest levels of government. Yeh (2021) drew upon multiple historical case examples to explain how and why even corrupt leaders may choose to embrace this type of powerful anticorruption reform. In many cases, leaders calculated that the short-term political benefit of embracing

reform outweighed their long-term calculations that those reforms might install courts that would convict them of corruption.

A localised understanding of everyday cultural practices in non-Western contexts is useful. However, even in Afghanistan, local actors understand that the existence of bribery, extortion, and grand corruption undermines the social contract between citizens and the state. Corruption alienates the majority of citizens who can ill-afford to devote a sizable slice of their household incomes to greedy police, corrupt judges, and government bureaucrats. While international norms regarding corruption may appear to be at odds with local, non-Western cultures, behaviour, and actions, local citizens are deeply dissatisfied with the existence of corruption, the burden of paying bribes, the failure of government to respond to their needs, and the seeming futility of fighting the system of corruption. The consequence is high levels of dissatisfaction with public services, protracted poverty, and distrust of the state.

The experiences of Hong Kong and Singapore demonstrate that international anti-corruption norms, legislation to fight corruption, and the establishment of independent inspectors to investigate corruption are not incompatible with Asian culture, mores, or behaviour. Neither Asian culture nor other cultures involving patronage, gift giving, *guanxi*, *blat*, or *bakhshish* are incompatible with the type of independent anticorruption inspectors and dedicated anticorruption courts that would be established by the APUN-CAC. The experiences of Hong Kong and Singapore demonstrate that citizens embrace clean government and prefer it to customary everyday practices that, in an earlier period, institutionalised reciprocal arrangements involving "gifts" in exchange for government services.

The APUNCAC is not incompatible with local cultural practices that involve patronage, gift giving, *guanxi*, *blat*, or *bakhshish.* The experience of Hong Kong demonstrates that the creation of an independent anticorruption agency, dedicated to independent investigations of corruption, can be effective despite the existence of culturally-engrained customs involving "tea money" and syndicated corruption in the police force. The creation of the ICAC was triggered by public outrage over the escape of Chief Superintendent Godber from Hong Kong. Citizens embraced the ICAC and the need to fight corruption. The establishment of the ICAC challenged the control of corrupt actors and was understood by citizens as a necessary, highly desirable response to corruption. Citizens who previously paid bribes because they could not fight the system were happy to lodge complaints about police corruption and appear as witnesses against corrupt police. The fact that citizens paid bribes was not an indicator that they accepted and condoned the tea money culture. They accurately understood that a lone citizen, absent the existence of independent anti-corruption inspectors, was simply outmatched and had no chance of fighting a corrupt system. In Hong Kong, public dissatisfaction and student protests resulted in the creation of a publicly trusted anticorruption agency and rapid reversal of the culture of corruption and impunity. The ICAC enjoyed strong political support from citizens, the British Crown administration, and successive governments, leading up to the handover of Hong Kong to China in 1997.

While it would be more challenging to curb customary patronage, personal relationships and the system of favours in Russia and China, the issue is not whether ordinary citizens support anticorruption reform, but whether the international community can assist reform-oriented elements in those countries to pressure political leaders to support the APUNCAC. The norms embedded in the APUNCAC are accepted by ordinary citizens. They are presumably rejected by corrupt actors, but if this was the sole determining factor, no law establishing strong investigators, strong prosecutors, and independent courts would ever be passed. Corrupt actors who resist reform exist in every nation, in every time period. That fact has not prevented reform. Corrupt actors pose a threat and slow reforms, but can—and have been—overcome in every nation that currently enjoys clean government. Yeh (2021) has described this process. It generally involves a process where support is assembled, by advocates for change, for a concrete piece of legislation that installs strong

measures to fight corruption. The APUNCAC is an example of international legislation that would install strong measures to fight corruption. The essential feature of the APUNCAC is the establishment of an independent body of inspectors modelled on Hong Kong's ICAC. The ICAC overcame powerful opposition, culturally-embedded, syndicated corruption in the police force, and a culture that seemingly accepted and embraced corruption as a permanent feature. There is no reason why this formula cannot be repeated elsewhere.

**Funding:** This research received no external funding.

**Institutional Review Board Statement:** Not applicable.

**Informed Consent Statement:** Not applicable.

**Data Availability Statement:** Not applicable.

**Conflicts of Interest:** The author declares no conflict of interest.

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
