# Peer review of "Anticorruption, Cultural Norms, and Implications for the APUNCAC"

_laws_

Round 1
Reviewer 1 Report
Interesting text with some outspoken points of vies concerning the policy on corruption. It is certainly worth to be published. However, I have some remarks: some concepts are not clear to me and need more explanation, the definition part and the Hong Kong part are both too long, the definition part lacks a coherent structure and I miss an explanation why the Asian culture is corruption-sensitive.
Here my remarks more in detail:
1. Concepts that need more explanation:
- a corruption complex
- moral economy
- a commission economy
- syndicated corruption
- folklore bureaucracy
2. Needs fundamental revision:
- the definition part:
- the definition part is a "mess" of all kind of definitions and descriptions. I don't see the difference in narrowly defined and a large definition and I miss a structure. Foremost all these definitions don't have an added value for the story. I suggest to rduce this chapter and to make some categories in order to make this chapter more convenient. eg. 1) low level corruption and high level corruption and 2) political and administrative corruption, 3) public corruption and private corruption. As the author says him or herself: typologies of corruption are helpful to identify several practices.
- inconsistency: you say that the political scientists sought to expand while Alemann is a political scientist .
- the Hong Kong part is far too long. The story of Godber is too detailed and leads to a loss of interest in the story. Try to shorten this part.
3. I wonder if the author can give an explanation why the Asian culture is susceptible to corruption. And why is it that corruption is something of rural communities? When I read it now it can be a story of a lot of more south European countries. What makes the difference between Asia and European countries on the lower stairs of the TI-ranking.
Reviewer 2 Report
It is a good read. The author(s) that in some instances corruption could be linked to some existing relationship with some official (lines 251 to 259) as noted in South Asia. This together with other points lamented does look like some form of systematic corruption embedded in the system. Despite the fact that examples of Hong Kong and Singapore could save as a model to follow, the article needs to address how the other countries can apply considering that countries have different dynamics and systems. Some questions are worth asking, Do other countries have in Asia and Africa have similar dynamics as Hong Kong and Singapore for the scenario of the latter to be applied across. You did acknowledge (as in the case of Hong Kong and Singapore) that corruption could be brought to the open and the domestic economies can apply the law according to your local jurisdiction, which is a good thing to take home. Being a descriptive article, it would have been nicer if your had statistics on the prevalence of corruption and the fight thereof as this could strengthen the article, plus also have the examples of Hong Kong and Singapore could be applied to mitigate the effects of systematic corruption. this is essential as we cannot ascertain whether or not political reforms could work for all states as systems are different.
